# Postoperative trachomatous trichiasis and associated factors among adults who underwent trachomatous trichiasis surgery in Ambassel District, North-East Ethiopia

**Abdu Tabor Yimam, Gizachew Tadesse Wassie◉\*, Getu Degu Alene**

Department of Epidemiology and Biostatistics, School of Public Health, Bahir Dar University, Bahir Dar, Ethiopia

\* leulgzat@gmail.com

## Abstract

**Data Availability Statement:** All relevant data are within the manuscript and its Supporting information files.

### Background

In Trachoma endemic countries, many people who underwent Trichiasis surgery faced a recurrence of the disease. Postoperative Trichiasis is a significant problem for patients and health care providers because it puts the eye at renewed risk of sight loss. Despite the low utilization of Trachomatous Trichiasis surgery and the high recurrence rate, evidence that elucidate why it recurs after surgery is limited. This study was aimed to assess the magnitude and associated factors of postoperative Trichiasis among 18 years and above individuals who underwent Trachomatous Trichiasis surgery between 2013 and 2019 in Ambassel District, Northeast Ethiopia, 2020.

### Methods

The community-based cross-sectional study design was conducted from March 10 to March 23/2020 in selected *kebeles* of Ambassel District. The required sample size (506) was calculated using EPI-INFO Version 7. A multi-stage sampling technique was used to employ study participants. Data were collected through the interviewer-administered structured pre-tested questionnaire and entered into EpiData version 3.1 and then exported to SPSS version 23.0 for analysis. Bi-variable and multivariable logistic regression models were fitted to identify associated factors of Postoperative Trachomatous Trichiasis.

### Results

Four hundred ninety two individuals participated in this study with a response rate of 97.2%. In Ambassel district, the prevalence of postoperative Trichiasis was 23.8% (95% CI = 19.9–27.8). Among associated factors of postoperative Trachomatous Trichiasis: age 50–59 (AOR = 3.34, CI = 1.38–8.1), 60–69 (AOR = 3.24, CI = 1.38–7.61), ≥70 years (AOR = 6.04, CI = 2.23–16.41), duration since surgery (AOR = 1.7, CI = 1.35–2.14), complication (AOR = 2.98, CI = 1.24–7.2), washing the face two times (AOR = 0.25, CI = 0.13–0.47), washing the face three and more times (AOR = 0.1, CI = 0.41–0.25), taking Azithromycin following

**Funding:** The author(s) received no specific funding for this work.

**Competing interests:** The authors have declared that no competing interests exist.

**Abbreviations:** AOR, Adjusted Odds Ratio; BTR, Bi-lamellar Tarsal Rotation; COR, Crude Odds Ratio; EFY, Ethiopian Fiscal Year; IECWs, Integrated Eye Care Workers; PLTR, Posterior Lamellar Tarsal Rotation; SAFE, Surgery Antibiotic Face cleanness Environmental improvement; SPSS, Statistical Product and service solution; TF, Trachomatous Inflammation, Follicular; TI, Trachomatous Inflammation, Intense; TPR, Tarsal Plate Rotation; TS, Trachomatous Scaring; TT, Trachomatous Trichiasis; TTC, Tetracycline; WHO, World Health Organization.

surgery (AOR = 0.19, CI = 0.09–0.41), pre-operative epilation history (AOR = 2.11, CI = 1.14, 3.9) and having a knowledge about TrachomaTtrichiasis (AOR = 0.21, CI = 0.08–0.58) showed a statistical significant association.

## Conclusions

The prevalence of postoperative Trichiasis in Ambassel District was higher than most Ethiopian studies. Age, frequency of face washing, medication following surgery, duration since the last surgery, knowledge about trachoma, pre-operative epilation history, and complication after surgery were identified to be independent factors. To minimize postoperative Trachomatous Trichiasis stakeholders need to consider health education for patients, provision of Azithromycin after surgery, and proper training for integrated eye care workers.

## Background

Trachomatous Trichiasis (TT) is a result of repeated infections of the inner part of the upper eyelid by Chlamydia Trachomatous and transmitted through contact with an infected person. It causes the upper eyelid to turn inwards so that eyelashes scrape on the eyeball–causing pain and overtime can result in scarring of the cornea [1]. If left unmanaged, TT can lead to visual impairment and blindness [2]. Trachomatous Trichiasis can be managed through surgery which involves the two most widely practiced surgical procedures; Bi-lamellar Tarsal Rotation (BTR) and Tarsal Plate Rotation (TPR/Tarsatomy) or Posterior Lamellar Tarsal Rotation (PLTR) as recommended by global elimination strategy led by the World Health Organization (WHO) [2].

Postoperative Trachomatous Trichiasis (POTT) is defined as when Trichiasis (one or more lashes touching the globe) is found again in an operated eye of a patient after surgery [2, 3]. Recurrence of Trichiasis following intervention is mentioned as a major barrier for the better uptake of TT surgery [3–5].

Globally, over 40 million people are estimated to suffer from active trachoma, and about 8.2 million have Trichiasis; 2.5 million people living with TT, 2.2 million have a visual impairment, and 1.2 million of these are irreversibly blinded from trachoma [6–8]. Trachomatous Trichiasis creates economic loss due to reduced productivity resulting from blindness and visual impairment [9].

In Ethiopia, 173,945 people received TT surgery and 39,339,311 received antibiotics through mass drug administration in 2017. In the Amhara region, 91,977 persons underwent the surgery, 13,651,377 doses of Azithromycin were distributed during mass drug administration; all villages received health education and 1,802,962 households have latrine facilities in 2017 [6].

Postoperative Trachomatous Trichiasis negatively impacts perception and uptake of surgery in communities. Individuals are less likely to seek surgery when they witness POTT in their community [10]. Additionally, re-operation has a higher risk of surgical failure than a primary operation and the eye may face the risk of blindness. Studies suggest that re-operation has up to a 13-fold increased risk of developing POTT [11].

Unfortunately, the rate of POTT after surgery is unacceptably high in many countries. A survey that assessed POTT in Africa found that the rate was ranged from 2.3% in 6 weeks to 65% in 7 years since surgery done[12, 13].

Despite the low utilization of TT surgery, there are limited researches that elucidate why TT recurs following surgery. Previous studies done in Ethiopia were few and in limited areas [14, 15]. Hence, this study was aimed to assess the magnitude of POTT and identifying its associated factors among adults who underwent TT surgery between 2013 and 2019 in the study area.

## Methods

This study was carried out in Ambassel district, South Wollo Administrative Zone, Amhara Region North East Ethiopia. According to the districts health office, the district provided TT surgery for 2,133 individuals between 2013 and 2019. An estimated 831 backlog TT cases were available in the district.

A community-based cross-sectional study design was conducted among residents of the selected *kebeles* of Ambassel district from March 10 to March 23, 2020. A total of 506 adult individuals who underwent TT surgery from 20013 to 2019 and living in selected *kebeles* of Ambassel district were included considering the assumptions of 95% CI, 1:1 cases to control ratio, 80% power, proportion of an outcome in unexposed group and adjusted odds ratio. Besides, 10 percent non-response rate and correction formula for small population size was applied.

### Sampling technique and procedures

A multistage sampling technique was used. In the district, from 24 total *kebeles* one urban *kebele* and seven rural *kebeles* were selected by the lottery method. A sampling frame was lists of individuals who underwent TT surgery and taken from the district office then, they accessed through their identification numbers given by community health information system. For each *kebele*, individuals with 18 years and above who underwent TT surgery were selected from their list using computer generated simple random sampling method. The number of study participants for each *Kebele* was determined by proportional allocation based on the number of individuals who had TT surgery. Repeated visits were made in situations when individuals who were supposed to be respondents were not present during the initial attempt.

### Data collection tools and procedures

Interviewer administered structured questionnaire was used to collect information on socio-demographic and household characteristics, knowledge about trachoma, medical and surgical factors and environmental factors from selected individuals. Knowledge was assessed using seven questions and categorizing above and below 50% as good knowledge and poor knowledge respectively. Latrine type also assessed whether it was constructed by simply obtained local materials, has a fully covered floor, wall, and roof, the well is long to have stood, the opening has cover and hand washing facility (improved) or not (unimproved). The questionnaire also contains an observational checklist containing variables such as presence or absence of inverted one or more eyelash, presence of Trachomatous infection, and surgeon's code who performed the procedure.

The data were collected by Integrated Eye Care Workers (IECWs) who had not performed surgery on any of the study communities and were blinded about which surgeon had conducted the surgery. The data collectors examined the eyes and recorded the findings. Ophthalmic loupe with a magnification of x2.5 and torchlight was used for this assessment. Also, checklist was used to collect information from TT surgery log book for variables like surgeon code, type of surgery, post-surgical treatment and eyelid operated. Participants who have any abnormalities diagnosed, the data collector referred the individual to the nearest health facility

for eye care according to national guidelines and 159 individuals who had active trachoma were provided Tetracycline eye ointment.

## Data quality assurance

To maintain the quality of the data, a pre-tested structured questionnaire was used to collect data. The pre-test was done in a nearby district rural community and an appropriate correction was taken to the questionnaire. Training was provided for the data collectors and supervisor for two days on how to conduct the interview, how to approach the respondents and perform the clinical examination. On-site supervision and feedback was given to data collectors. Data entry was done by the principal investigator after checking for completeness and coding.

## Data processing and analysis

The collected data were checked for completeness and entered into EpiData version 3.1 software programs and then exported to SPSS version 23 software programs for data cleaning and analysis. Bi-variable logistic analysis was performed to see an association of each independent variable with the POTT. Explanatory variables with P-value $\leq$ 0.2 were selected as candidates for the final model. Binary logistic regression model (multivariable analysis) was performed to assess the association between the independent variables and the outcome variable. The level of significance was set at 95% CI (p < 0.05) for statistical significance in the model. To assess the model fitness for the application of multivariable logistic regressions, Hosmer–Lemeshow goodness-of-fit test statistic was employed and it was p-value = 0.975.

**Ethics approval and consent to participate.** Ethical approval was obtained from the College of Medical and health science Institutional Review Board (IRB) Committee (Protocol number: 00134/2020), of Bahir Dar University. Written informed consent was obtained from each study participant. The study participants were also provided information about the objectives and expected outcomes of the study. Information obtained from individual participants was kept secure and confidential.

## Results

### Socio-demographic characteristics of study participants

The response rate of the study was 97.2%. Of the non-respondents, two of them were unable to respond due to illness, three were not present after three times visit, seven of them left to other districts and the remaining two died due to other causes. The age distribution of the study participants showed that about 36% of them were below 50 years of age. The mean age of the study participants was 55.03 (SD ±11.13) years. Among the total respondents, 60.6% were females, 74.2% were married, 44% have had at least one child in the household, 93% were farmers, and 80.5% did not have formal education (Table 1).

### Personal and environmental characteristics of study participants

Forty three percent of study participants reported that they washed their face two times a day, 71% did not use soap when washing their faces at least once per day. Nearly half of respondents did not have toilet facilities; among 264 individuals who have had toilet facilities, only 6% had improved ones (Table 2).

**Table 1. Socio-demographic characteristics of respondents in Ambassel District, South Wollo Zone, Northeast Ethiopia, 2020.**

| Variables | | Frequency | % |
|---|---|---|---|
| Age in years | <50 | 175 | 35.6 |
| | 50–59 | 123 | 25 |
| | 60–69 | 147 | 29.9 |
| | ≥70 | 47 | 9.6 |
| Sex | Male | 194 | 39.4 |
| | Female | 298 | 60.6 |
| Marital status | Single | 10 | 2.0 |
| | Married | 365 | 74.2 |
| | Divorced | 42 | 8.5 |
| | Widowed | 75 | 15.2 |
| Responsibility in house | Head | 446 | 90.7 |
| | Other family members | 46 | 9.3 |
| Family size | Family size <4 | 189 | 38.4 |
| | Family size > = 4 | 303 | 61.6 |
| <10 years old children in the household | Yes | 217 | 44.1 |
| | No | 275 | 55.9 |
| Occupation | Farmer | 457 | 92.9 |
| | Merchant | 35 | 7.1 |
| Education status | No formal Education | 396 | 80.5 |
| | Elementary | 96 | 19.5 |
| Residence | Urban | 82 | 16.7 |
| | Rural | 410 | 83.3 |
| Religion | Orthodox | 270 | 54.9 |
| | Muslim | 210 | 42.7 |
| | Protestant | 12 | 2.4 |

(N = 492)

## Medical and surgical characteristics of respondents

Sixty percent of participants, received the service at the health center, 36.2% at the health post, and the remaining 3.9% served at the hospital. Nearly, thirteen percent had performed epilation after surgery. Ninety eight percent of respondents received the annual routine Zithromax campaign medication and about 92% of the study participants' families took this routine Zithromax campaign medication (Table 3).

## The prevalence of postoperative trachomatous trichiasis

In this study, the prevalence of POTT was 23.8% (95% CI: 19.9–27.8). The majority of recurrence occurred in a single eye which was 48% right eye, 47% left eye, and 5% in both eyes.

## Factors associated with postoperative trachomatous trichiasis

The bi-variable binary logistic regression showed that sex, age, marital status, responsibility in the household, occupation, education status, frequency of face washing, soap use at least once a day, availability of toilet, travel time to get water access, the number of surgeries, place of surgery, operated eye type, medication following surgery, duration since last surgery, pre-operative epilation history, complication after surgery, and knowledge about trachoma/Trichiasis

**Table 2. Personal and environmental characteristics of respondents in Ambassel District, South Wollo, Northeast Ethiopia, 2020.**

| Variables | | Frequency | % |
|---|---|---|---|
| Frequency of face washing | Once a day | 137 | 27.8 |
| | Twice a day | 213 | 43.3 |
| | Three and above per day | 142 | 28.9 |
| Soap use | Yes | 142 | 28.9 |
| | No | 350 | 71.1 |
| Knowledge about trachoma | Good | 434 | 88.2 |
| | Poor | 35 | 7.1 |
| Have a toilet | Yes | 264 | 53.7 |
| | No | 228 | 46.3 |
| Type of toilet | Improved | 16 | 6.1 |
| | Unimproved | 248 | 93.9 |
| Toilet utilization | Yes | 251 | 95.1 |
| | Some of the family | 13 | 4.9 |
| Travel time to get water | <30 minute | 235 | 47.8 |
| | >30 minute | 257 | 52.2 |

(N = 492)

were statistically significant association with POTT at p-value < 0.2. These variables were candidates for the multivariable analysis.

Multiple binary logistic regressions (multivariable analysis) showed that age, frequency of face washing, duration since last surgery, medication following surgery, complication after surgery, pre-operation epilation history, and knowledge about trachoma/Trichiasis were factors associated with POTT, at P < 0.05 with 95% CI.

Participants aged 50 to 59 years were 3.34 times (AOR = 3.34, 95%CI: 1.38, 8.1), 60 and 69 years were 3.24 times (AOR = 3.24, 95% CI: 1.38, 7.61), and ≥ 70 years were six times (AOR = 6.04, 95% CI: 2.23, 16.41) more likely to develop POTT compared to those who were < 50 years old respectively.

An individual who washed his/her face twice a day was 75% (AOR = 0.25, 95% CI: 0.13, 0.47) less likely to develop POTT compared with the one washing his/her face once a day. Similarly, an individual who washed his/her face three and above times per day was 90% less likely (AOR = 0.10, 95% CI: 0.04, 0.25) to develop POTT compared with the same reference group.

Those who had taken Azithromycin following surgery were 81% (AOR = 0.19, 95% CI: 0.09, 0.41) less likely to have POTT than those who took Tetracycline eye ointment. When the duration since last surgery increased by one year, POTT was 1.7 times more likely to occur keeping other variables constant (AOR = 1.7, 95% CI: 1.35, 2.14).

Respondents who had complications after surgery were three times (AOR = 2.98, 95% CI: 1.24, 7.2) more likely to have had POTT compared with those who did not had complication keeping other variables constant. Individuals who had pre-operative epilation history were two times (AOR = 2.11, 95% CI: 1.14, 3.9) more likely to develop POTT compared with those who did not have history of pre-operative epilation keeping other variables constant (Table 4).

## Discussion

The current study revealed that the prevalence of POTT was 23.8%. The result was consistent with the studies conducted in the Amhara Region, Ethiopia, Nepal and Tanzania; where

**Table 3. Medical and surgical characteristics of study participants in Ambassel District, South Wollo Zone, Northeast Ethiopia, 2020.**

| Variables | | Frequency | % |
|---|---|---|---|
| Number of surgery | One time | 409 | 83.1 |
| | Two and above times | 83 | 16.9 |
| Eyelid operated | Right | 131 | 26.6 |
| | Left | 156 | 31.7 |
| | Both | 205 | 41.7 |
| Medication type following surgery | Azithromycin/oral tablet | 173 | 35.2 |
| | Tetracycline eye ointment | 319 | 64.8 |
| Duration since last surgery in years | One year | 39 | 7.9 |
| | Two years | 134 | 27.2 |
| | Three years | 137 | 27.8 |
| | Four years | 81 | 16.5 |
| | Five years | 60 | 12.2 |
| | Six years | 41 | 8.3 |
| History of epilation before surgery | Yes | 150 | 30.5 |
| | No | 342 | 69.5 |
| Frequency of epilation before surgery | once a week | 52 | 10.6 |
| | once per two week | 48 | 9.8 |
| | once per month | 50 | 10.2 |
| Complications after surgery | Yes | 46 | 9.3 |
| | No | 446 | 90.7 |
| Surgeons code who perform this operation | A | 72 | 14.6 |
| | B | 94 | 19.1 |
| | C | 74 | 15.0 |
| | D | 74 | 15.0 |
| | E | 101 | 20.5 |
| | F | 61 | 12.4 |
| | Other hospital nurses | 16 | 3.3 |
| Face cleanliness | Yes | 421 | 86.4 |
| | No | 71 | 14.4 |
| Current trachoma status | Yes | 154 | 31.3 |
| | No | 338 | 68.7 |

(N = 492)

POTT rate was 24.7%, 25%, and 27.9% respectively [2, 15, 16]. However, it is lower than other studies done in Tanzania where the POTT rate was 31%, Gambia 41%, Oman 47.2%, and Nepal 28.9% [4, 17–19]. This difference might be due to the differences in surgical procedures. The study of Tanzania, Nepal, and Vietnam were used other than the PLTR technique. A study in the West Gojam Zone, Ethiopia shows PLTR procedure has a lower recurrence rate [20]. Also, the finding of this study was higher than most studies conducted in Ethiopia ranges from 8% to 19% [14, 21–25], the Jigawa State of Nigeria 17.3% [26], and Vietnam 10.8% to 15.9% [27, 28]. This difference might be because the small sample size of the study and differences in the study population in which previous studies excluded individuals with repeat surgery that might reduce the POTT.

The elderly age had a significant association with TT recurrence. As people get older, the likely hood of having POTT increases by threefold. This finding was similar to the studies conducted in West Gojjam Zone, Ethiopia [15, 20], and in four districts of Vietnam [27]. This

**Table 4. Multi-variable analysis of binary logistic regression model for various factors and postoperative Trichiasis in Ambassel District, South Wollo Zone, Northeast Ethiopia, 2020.**

| Variables | | Postoperative outcome | | COR(95%CI) | AOR(95%CI) | P-value |
|---|---|---|---|---|---|---|
| | | Recurrence | No recurrence | | | |
| Age | <50 | 11 | 164 | 1.00 | 1.00 | |
| | 50–59 | 23 | 100 | 3.43(1.6–7.34) | 3.34(1.38–8.1) | 0.007 |
| | 60–69 | 57 | 90 | 9.44(4.71–18.92) | 3.24(1.38–7.61) | 0.007 |
| | > = 70 | 26 | 21 | 18.46(7.98–42.7) | 6.04(2.25–16.41) | <0.000 |
| Sex of Respondent | Male | 40 | 154 | 0.75(0.48–1.15) | 0.72(0.38–1.36) | 0.312 |
| | Female | 77 | 221 | 1 | 1 | |
| marital status | Single | 0 | 10 | 0 | 0 | .999 |
| | Married | 73 | 292 | 0.27(0.16–0.46) | 1.34(0.59–3.04) | .488 |
| | Divorced | 8 | 34 | 0.5(0.11–0.62) | 0.5(0.13–1.85) | .297 |
| | Widowed | 36 | 39 | 1 | 1 | |
| Responsibility in the house | Head | 98 | 348 | 0.4(0.22–0.75) | 1.15(0.38–3.49) | 0.807 |
| | Other family member | 19 | 27 | 1 | 1 | |
| Occupation of the respondent | Farmer | 112 | 345 | 1.95(0.74–5.14) | 0.83(0.18–3.84) | 0.815 |
| | Merchant | 5 | 30 | 1 | 1 | |
| Educational status | No formal Education | 107 | 289 | 3.18(1.6–6.36) | 1.45(0.48–4.37) | 0.506 |
| | Elementary | 10 | 86 | 1 | 1 | |
| Frequency of face washing | Once a day | 68 | 69 | 1 | 1 | |
| | Twice a day | 37 | 176 | 0.22(0.13–0.35) | 0.25(0.13–0.47) | <0.000 |
| | Three and above per day | 12 | 130 | 0.1(0.05–0.19) | 0.1(0.04–0.25) | <0.000 |
| Soap use | Yes | 21 | 121 | 0.46(0.27–0.77) | 0.76(0.36–1.59) | 0.469 |
| | No | 96 | 254 | 1 | 1 | |
| Have a toilet | Yes | 46 | 217 | 0.47(0.31–072) | 0.76(0.4–1.45) | 0.409 |
| | No | 71 | 158 | 1 | 1 | |
| Travel to get water | <30 minute | 50 | 185 | 0.77(0.50–1.17) | 0.94(0.5–1.75) | 0.835 |
| | >30 minute | 67 | 190 | 1 | 1 | |
| Number of surgery | One time | 80 | 329 | 0.30(0.18–0.50) | 1.31(0.64–2.65) | 0.462 |
| | Two and more times | 37 | 46 | 1 | 1 | |
| Place of surgery for the last time | Health post | 46 | 132 | 1 | 1 | |
| | Health Centre | 70 | 225 | 0.89(0.58–1.37) | 0.72(0.39–1.3) | 0.288 |
| | Hospital | 1 | 18 | 0.16(0.02–1.23) | 0.13(0.01–1.35) | 0.088 |
| eyelid operated | Right | 25 | 106 | 0.43(0.25–0.72) | 1.03(0.49–2.15) | 0.940 |
| | Left | 19 | 137 | 0.25(0.14–0.44) | 0.56(0.26–1.21) | 0.142 |
| | Both | 73 | 132 | 1 | 1 | |
| Medication type following surgery | Azithromycin/ oral tablet | 18 | 155 | 0.26(0.15–0.44) | 0.19(0.09–0.41) | <0.000 |
| | Tetracycline eye ointment | 99 | 220 | 1 | 1 | |
| Duration of last surgery in years, median | | 3 | | 1.53(1.31–1.78) | 1.7(1.35–2.14) | <0.000 |
| History of epilation before surgery | Yes | 70 | 80 | 5.49(3.52–8.57) | 2.11(1.14–3.9) | 0.017 |
| | No | 47 | 295 | 1 | 1 | |
| Complications after surgery | Yes | 25 | 21 | 4.58(2.46–8.55) | 2.98(1.24–7.2) | 0.015 |
| | No | 92 | 354 | 1 | 1 | |
| Knowledge status category | Good | 80 | 354 | 0.08(0.04–0.17) | 0.21(0.08–0.58) | 0.008 |
| | Poor | 26 | 9 | 1 | 1 | |

might be because as the age gets older, the body's immune response may be decreased [29]. Additionally, the elderly usually have the highest prevalence of trachoma that can be attributed to additional contracture of the tarsal tissues leading to recurrence of Trichiasis.

In this study, increased frequency of the face washing was statistically significant preventive factor of POTT. Though face washing is one component of the WHO SAFE strategy to eliminate trachoma with direct relation to reducing Trichiasis recurrence, no previous study had investigated the association between frequency of face washing and Trichiasis recurrence. The association might be because individuals who wash their faces would less frequently be affected by trachoma which is a measure cause of TT recurrence.

Medication following TT surgery was the factor that has a significant association with the outcome variable. In this study, Azithromycin was associated with significantly less POTT rate compared with Tetracycline. This result was similar to the study done in Wolayta Soddo, Ethiopia [30] and Nepal [17]. But another randomized controlled trial study conducted in the Gambia shows that Azithromycin did not improve the outcome of Trichiasis surgery [31]. The possible reason might be a difference in the study participants who received medication following surgery. In the above Gambian study, all patients received Tetracycline eye ointment twice a day for two weeks duration, and subject randomized to Azithromycin group also received a one gram dose of Azithromycin. But in our study, the participants were taken either Tetracycline or Azithromycin. Treatment with Azithromycin is an important component of the WHO SAFE strategy to eliminate trachoma by breaking the cycle and spread of ocular infection.

This study shows that when the duration since the last surgery was longer; the risk of POTT was increased. It was in line with two studies conducted in Oman that found the length of time since surgery was an important predictor of Trichiasis recurrence after surgery. In those studies, during the early postoperative period the recurrence was not significantly high [18, 32]. As the duration increased the possibility of an individual having POTT might increase due to the progression of the cicatricial process. But it contradicted with the study conducted in rural *kebeles* of west Gojam Zone, Ethiopia which shows that when the time is long since surgery, the risk of recurrence was decreased [15]. This difference might be due to our study includes individuals who have up to six years of duration.

When an individual has a good knowledge towards Trachoma and Trichiasis, the odds of having POTT were decreased significantly. Good knowledge might result a positive effect on the prevention of trachoma infection. The study conducted in Alaba District, Southern Ethiopia reported the negative effect of poor knowledge on Trachoma prevention practices [33]. Increasing the community knowledge about Trachoma is helpful to eliminate blinding due to Trachoma locally as well as regionally.

Complication experience after TT surgery was another factor that increases the risk of a POTT. This finding was in line with the study conducted in rural villages of West Gojjam Zone, Ethiopia [15].

Individuals who had pre-operative epilation history as a self-treatment practice were found to be at higher odds of having POTT. This finding was in line with the study conducted in Wolayta zone, Southern Ethiopia [22]. But another study in a similar study area was found epilation before surgery had no association with POTT [34]. This difference might be because epilation history was self-reported and it might lead to misclassification error. Repetitive epilation practice might harm the matrix, the portion of the hair that grows actively, and follicular stem cells. Also, it makes the new hair to be sharp, and the epidermis becomes less responsive possibly due to a decrease in the number of stimulus-responsive cells.

### Limitations of the study

The study could not include some factors like pre-operative TT severity, incision length, and suture type which were impossible to find the data.

The other limitation was related to the development of Trichiasis. It was not known exactly when the individual developed Trichiasis since recurrence in the early postoperative period may occur for different reasons than late recurrence.

## Conclusions

The prevalence rate of POTT in the study area was higher than most Ethiopian studies. Our finding indicated that the risk of POTT increased due to old age, long duration since surgery, pre-operative epilation history and complication after surgery. On the other hand, face washing more than one time per day, good knowledge about Trachoma/Trichiasis and taking Azithromycin medication following surgery decreased the rate of POTT.

Improving the knowledge, personal hygiene and environmental sanitation in the community, establishing Trachoma screening program for older adults, applying Azithromycin medication for all patients, implementing follow up visits to patients who have had TT surgery, and management of complications are important measures that need to be taken to decrease the rate of POTT.

## Supporting information

**S1 Dataset.**
(XLSX)

## Acknowledgments

The authors would like to thank, Bahir Dar University, the study participants who freely give their time and Amhara health bureau and Ambassel district health office for facilitating the study and their support.

## Author Contributions

**Conceptualization:** Abdu Tabor Yimam.

**Data curation:** Abdu Tabor Yimam.

**Formal analysis:** Abdu Tabor Yimam.

**Methodology:** Abdu Tabor Yimam, Gizachew Tadesse Wassie, Getu Degu Alene.

**Software:** Abdu Tabor Yimam, Gizachew Tadesse Wassie, Getu Degu Alene.

**Supervision:** Gizachew Tadesse Wassie, Getu Degu Alene.

**Validation:** Getu Degu Alene.

**Writing – original draft:** Abdu Tabor Yimam, Gizachew Tadesse Wassie.

**Writing – review & editing:** Gizachew Tadesse Wassie, Getu Degu Alene.

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
