## [Decision Letter · Decision Letter 0]

26 Dec 2023

PONE-D-23-08111Postoperative Trichiasis  and Associated Factors Among Adults who underwent Trachomatous Trichiasis Surgery in Ambassel District, North-East Ethiopia.PLOS ONE

Dear Dr. Wassie,

Thank you for submitting your manuscript to PLOS ONE. After careful consideration, we feel that it has merit but does not fully meet PLOS ONE’s publication criteria as it currently stands. Therefore, we invite you to submit a revised version of the manuscript that addresses the points raised during the review process.

We look forward to receiving your revised manuscript.

Kind regards,

Mesfin Gebrehiwot Damtew (PhD)

Academic Editor

PLOS ONE

Journal Requirements:

Reviewers' comments:

Reviewer's Responses to Questions

**Comments to the Author**

1. Is the manuscript technically sound, and do the data support the conclusions?

Reviewer #1: Partly

Reviewer #2: Yes

2. Has the statistical analysis been performed appropriately and rigorously? 

Reviewer #1: I Don't Know

Reviewer #2: Yes

3. Have the authors made all data underlying the findings in their manuscript fully available?

Reviewer #1: Yes

Reviewer #2: Yes

4. Is the manuscript presented in an intelligible fashion and written in standard English?

Reviewer #1: No

Reviewer #2: No

5. Review Comments to the Author

Reviewer #1: This paper describes a study in Ambassel District in Ethiopia where a regression was performed to determine if post-operative TT is correlated with several variables. The results demonstrate a correlation between post-operative TT and patient age, duration since surgery, epilation history, face washing, knowledge about trachoma, and post surgery azithromycin. This information is helpful to the community to inform resource alignment for supporting TT surgery. The study appears to have been designed and performed appropriately. However, the writing in this manuscript is not sufficient for publication at this time. A thorough copy edit is needed and then I would be very happy to re-review.

It is my understanding that this study was focused on trachomatous trichiasis rather than all cause trichiasis. If this assumption is correct, I recommend the following copy edits.

- Reword first paragraph to include background on what TT is and how it’s addressed (trachoma is the world’s leading cause of preventable blindness…morbidity from trachoma is called trachomatous trichiasis (TT)… TT can be managed through surgery … details of what that surgery entails…etc.)

- If you include this information about what trachoma is and where TT comes into play in the natural progression of the disease then you no longer need the paragraph about the WHO grading system.

- Why is the wealth index included as a figure – this information does not contribute to the study and so does not warrant a figure.

- The details on the status of implementation in Ethiopia should come after the paragraph on the global situation. Additionally, the Ethiopia sentence is unclear and needs a copy edit.

The methods sections read like a protocol rather than a manuscript. Please consolidate the language into one clear Methods section. For example, the sample size section could be one sentence in your methods where you describe the power and confidence of your sample size. Remove variables and operational definitions sections.

The analysis is generally appropriate. However, I’m concerned about confounding among some of the variables. I am also not convinced that martial status, responsibility in the household, and occupation should be included in the analysis as the sample is not balanced for these categories. Marital status is 74% married and 15% widowed (which may be confounded with age). Responsibility in the house was 90% head. Occupation was 93% farmer.

The claims in this paper align with previous literature and the authors demonstrate this consistency in the discussion. However, there is need for tightening the writing of this section and drawing the “so what” out a bit more.

Specific copy edits:

- Please use consistent capitalization in the title of this paper

- Consistently use either trichiasis or trachomatous trichiasis

- When providing data do not use terms like “nearly” or “about” if the value is 92% then just say that.

- Write out the word “percent” when using it on its own in a sentence rather than using %.

- Table 3. Remove “surgeons code” variables – unless you ran an analysis to see if there was a correlation between a particular surgeon and post-operative TT. If you did this, then please include in the text of the manuscript.

Select more meaningful Key Words

Reviewer #2: First and foremost, I'd like to thank the authors of this research for their contribution to Assessing the Postoperative Trichiasis and Associated Factors among Adults who underwent Trachomatous Trichiasis Surgery in Ambassel District, North-East Ethiopia. Herewith my comments and suggestions on each of the manuscript sections

1. Title

Seems good

2. Introduction

• I didn't comprehend the research's novelty. By properly demonstrating why this research was centered on this title, the authors of this manuscript should anticipate to convix both outsiders and insiders of the subjects. In short, the uniqueness of the study should be highlighted.

3. Method section

• The author of this manuscript should also explain how validity and reliability were ensured.

• How are the effects of confiding variables managed?

4. Results

• Good looking!

5. Discussion

• First, the author should clearly interpret what the results mean. Second, compare the findings with comparable populations and settings (attention to the level of heath care facility included). Third, state the possible explanation for the disparity between this study's findings and prior literature after careful review. Fourth, the possible public health implications should also be stated in the discussion section. The author should also state the strengths and limitations of the study.

• Firth, methodological discussion should be recommended (Attention to internal and external validity).

6. Conclusions and recommendations.

• The concussions section seems what expected from discussion section. For instances: The prevalence rate of postoperative Trichiasis in the study area was higher than most Ethiopian studies.

• The recommendation seems based on general facts. So, make a sound, actionable, and conclusion-based recommendation after revising the conclusion section.

7. References

• Should be relevant to the population and setting of this investigation.

6. PLOS authors have the option to publish the peer review history of their article (what does this mean?). If published, this will include your full peer review and any attached files.

Reviewer #1: No

Reviewer #2: No

---

## [Author Response · Author response to Decision Letter 0]

19 Jan 2024

Response to the Reviewers

Dear editor, thank you for considering our manuscript for publication; we have revised the contents based on the reviewer’s comments as below.

Ref: Submission ID : PONE-D-23-08111

Title: "Postoperative Trachomatous Trichiasis and Associated Factors Among Adults who underwent TT Surgery in Ambassel District, North-East Ethiopia”.

PLOS ONE."

Reviewers' comments:

Reviewer's Responses to Questions

Comments to the Author

1. Is the manuscript technically sound, and do the data support the conclusions?

Reviewer #1: Partly

Reviewer #2: Yes

2. Has the statistical analysis been performed appropriately and rigorously?

Reviewer #1: I Don't Know

Reviewer #2: Yes

3. Have the authors made all data underlying the findings in their manuscript fully available?

Reviewer #1: Yes

Reviewer #2: Yes

4. Is the manuscript presented in an intelligible fashion and written in standard English?

Reviewer #1: No

Reviewer #2: No

5. Review Comments to the Author

Journal Requirements:

Response: We followed all journals requirements.

Response: We have uploaded as supportive document files.

Response: we moved the ethics statement from declaration section to method section (page #11)

Reviewer #1: This paper describes a study in Ambassel District in Ethiopia where a regression was performed to determine if post-operative TT is correlated with several variables. The results demonstrate a correlation between post-operative TT and patient age, duration since surgery, epilation history, face washing, knowledge about trachoma, and post surgery azithromycin. This information is helpful to the community to inform resource alignment for supporting TT surgery. The study appears to have been designed and performed appropriately. However, the writing in this manuscript is not sufficient for publication at this time. A thorough copy edit is needed and then I would be very happy to re-review.

It is my understanding that this study was focused on trachomatous trichiasis rather than all cause trichiasis. If this assumption is correct, I recommend the following copy edits.

- Reword first paragraph to include background on what TT is and how it’s addressed (trachoma is the world’s leading cause of preventable blindness…morbidity from trachoma is called trachomatous trichiasis (TT)… TT can be managed through surgery … details of what that surgery entails…etc.)

- If you include this information about what trachoma is and where TT comes into play in the natural progression of the disease then you no longer need the paragraph about the WHO grading system.

Response: Thank you for your feedback. We have revised as the reviewer’s suggestions(lines#55-75).

- Why is the wealth index included as a figure – this information does not contribute to the study and so does not warrant a figure.

Response: Thank you for the feedback. We removed figure 1 and its descriptions (page#11, lines 246-247).

- The details on the status of implementation in Ethiopia should come after the paragraph on the global situation. Additionally, the Ethiopia sentence is unclear and needs a copy edit.

Response: Thank you for the feedback. We revised as your recommendations.

The methods sections read like a protocol rather than a manuscript. Please consolidate the language into one clear Methods section. For example, the sample size section could be one sentence in your methods where you describe the power and confidence of your sample size. Remove variables and operational definitions sections.

Response: We revised it.

The analysis is generally appropriate. However, I’m concerned about confounding among some of the variables. I am also not convinced that martial status, responsibility in the household, and occupation should be included in the analysis as the sample is not balanced for these categories. Marital status is 74% married and 15% widowed (which may be confounded with age). Responsibility in the house was 90% head. Occupation was 93% farmer.

Response: We acknowledge the reviewer’s concern. However, we believe that these potential confounding variables would be managed in the analysis stage; since we conducted multivariable logistic regression model. 

The claims in this paper align with previous literature and the authors demonstrate this consistency in the discussion. However, there is need for tightening the writing of this section and drawing the “so what” out a bit more.

Response: Thank you for the feedback. We tried to show the clinical and public health implications the results in the discussion and conclusion sections.

Specific copy edits:

- Please use consistent capitalization in the title of this paper

- Consistently use either trichiasis or trachomatous trichiasis

Response: Thank you for the feedback. We used Trachomatous Trichiasis throughout the document.

- When providing data do not use terms like “nearly” or “about” if the value is 92% then just say that.

- Write out the word “percent” when using it on its own in a sentence rather than using %.

Response: Thank you for the feedback. We followed your suggestions.

- Table 3. Remove “surgeons code” variables – unless you ran an analysis to see if there was a correlation between a particular surgeon and post-operative TT. If you did this, then please include in the text of the manuscript.

Response: We have analyzed this variable and there was no association between surgeon code and the outcome variable.

Select more meaningful Key Words

Response: Thank you for your feedback. We revised key words.

Reviewer #2: First and foremost, I'd like to thank the authors of this research for their contribution to Assessing the Postoperative Trichiasis and Associated Factors among Adults who underwent Trachomatous Trichiasis Surgery in Ambassel District, North-East Ethiopia. Herewith my comments and suggestions on each of the manuscript sections

1. Title

Seems good

2. Introduction

• I didn't comprehend the research's novelty. By properly demonstrating why this research was centered on this title, the authors of this manuscript should anticipate to convix both outsiders and insiders of the subjects. In short, the uniqueness of the study should be highlighted.

Response: thank you for your feedback. We revised the introduction section to address the reviewer’s concerns.

3. Method section

• The author of this manuscript should also explain how validity and reliability were ensured.

Response: the validity and reliability of the study was insured through following standard procedures and using pretested data collection tools.

• How are the effects of confiding variables managed?

Response: We believe that potential confounding variables would be managed in multivariable logistic regression analysis.

4. Results

• Good looking!

Response: thank you for your acknowledgment.

5. Discussion

• First, the author should clearly interpret what the results mean. Second, compare the findings with comparable populations and settings (attention to the level of heath care facility included). Third, state the possible explanation for the disparity between this study's findings and prior literature after careful review. Fourth, the possible public health implications should also be stated in the discussion section. The author should also state the strengths and limitations of the study.

• Firth, methodological discussion should be recommended (Attention to internal and external validity).

Response: Thank you for your feedback and suggestions. We revised the whole discussion section to address the reviewer’s concerns.

6. Conclusions and recommendations.

• The concussions section seems what expected from discussion section. For instances: The prevalence rate of postoperative Trichiasis in the study area was higher than most Ethiopian studies.

• The recommendation seems based on general facts. So, make a sound, actionable, and conclusion-based recommendation after revising the conclusion section.

Response: We revised conclusion and recommendation sections in line with the objective of the study and our findings.

7. References

• Should be relevant to the population and setting of this investigation.

Response: Thank you for your feedback.

---

## [Decision Letter · Decision Letter 1]

22 Feb 2024

PONE-D-23-08111R1Postoperative Trachomatous Trichiasis  and Associated Factors Among Adults who underwent Trachomatous Trichiasis Surgery in Ambassel District, North-East Ethiopia.PLOS ONE

Dear Dr. Wassie,

Thank you for submitting your manuscript to PLOS ONE. After careful consideration, we feel that it has merit but does not fully meet PLOS ONE’s publication criteria as it currently stands. Therefore, we invite you to submit a revised version of the manuscript that addresses the points raised during the review process.

We look forward to receiving your revised manuscript.

Kind regards,

Mesfin Gebrehiwot Damtew (PhD)

Academic Editor

PLOS ONE

Reviewers' comments:

Reviewer's Responses to Questions

**Comments to the Author**

1. If the authors have adequately addressed your comments raised in a previous round of review and you feel that this manuscript is now acceptable for publication, you may indicate that here to bypass the “Comments to the Author” section, enter your conflict of interest statement in the “Confidential to Editor” section, and submit your "Accept" recommendation.

Reviewer #1: (No Response)

2. Is the manuscript technically sound, and do the data support the conclusions?

Reviewer #1: Partly

3. Has the statistical analysis been performed appropriately and rigorously? 

Reviewer #1: Yes

4. Have the authors made all data underlying the findings in their manuscript fully available?

Reviewer #1: Yes

5. Is the manuscript presented in an intelligible fashion and written in standard English?

Reviewer #1: No

6. Review Comments to the Author

Reviewer #1: I appreciate the author’s hard work on this study and believe it’s important for this information to be available for the global community. However, the paper still requires significant copy-editing. I have provided some initial suggested edits below, but the full paper would greatly benefit from a full copy edit by someone with experience publishing in an English language journal.

Since you abbreviate trachomatous trichiasis as TT early in the paper, continue using TT throughout rather than spelling out the full name.

Same comment on using POTT throughout.

Line 56 – rather than stating “…transmitted through contact with eye discharge” I recommend a broader statement, such as “…transmitted through contact with an infected person”. This is because the bacteria could be transmitted via nasal discharge or other fomites. (Trachoma - StatPearls - NCBI Bookshelf (nih.gov))

Line 58 – change “resulting in excruciating pain….” To say something like “causing pain and overtime can result in scarring of the cornea”. The large spectrum of clinical manifestations of TT means that some people do not develop scarring, but many do. So better for this language to reflect this. You all should cite this statement.

Line 59 – same comment at line 58 about the large spectrum of clinical manifestations. I would say “If left unmanaged TT can lead to visual impairment and blindness.” You all should cite this statement.

Line 60 – Trachoma is the leading cause of preventable blindness. I don’t think this is the correct citation. Please double check.

Line 61 – TT can be managed through surgery which involves….(briefly describe the procedure and cite the WHO manual)

Line 68 – what is “the plan”? since you give a true value “173,945” remove “about”. I think what you are saying is “In Ethiopia, 173,945 received TT surgery and 39,339,311 people received antibiotics through mass drug administration in 2017.”

Line 71 – all villages received health education….

Line 75/76 – suggest rewording “TT creates economic loss due to reduced productivity resulting form blindness and visual impairment.”

Line 77 – POTT negatively impacts perception and uptake of surgery in communities. Individuals are less likely to seek surgery when they witness POTT in their community. Additionally, re-operation has…..”

Line 85 – POTT

Line 88/89 – confusing wording

Line 219 – first time to mention PLTR – I suggest mentioning this approach in the background and you also need to spell out the acronym the first time you use it.

Aside from copy-edit. Could the authors please explain how they accounted for quality of original surgery? Were all the surgeons in the study of a certain caliber?

7. PLOS authors have the option to publish the peer review history of their article (what does this mean?). If published, this will include your full peer review and any attached files.

Reviewer #1: No

---

## [Author Response · Author response to Decision Letter 1]

10 Apr 2024

Response to the Reviewers

Dear editor, thank you for considering our manuscript for publication; we have revised the contents based on the reviewer’s comments as below.

Ref: Submission ID: PONE-D-23-08111R1

Title: Postoperative Trachomatous Trichiasis and Associated Factors among Adults who underwent Trachomatous Trichiasis Surgery in Ambassel District, North-East Ethiopia.

PLOS ONE

Reviewers' comments:

Reviewer's Responses to Questions

Comments to the Author

1. If the authors have adequately addressed your comments raised in a previous round of review and you feel that this manuscript is now acceptable for publication, you may indicate that here to bypass the “Comments to the Author” section, enter your conflict of interest statement in the “Confidential to Editor” section, and submit your "Accept" recommendation.

Reviewer #1: (No Response)

2. Is the manuscript technically sound, and do the data support the conclusions?

Reviewer #1: Partly

3. Has the statistical analysis been performed appropriately and rigorously?

Reviewer #1: Yes

4. Have the authors made all data underlying the findings in their manuscript fully available?

Reviewer #1: Yes

5. Is the manuscript presented in an intelligible fashion and written in standard English?

Reviewer #1: No

6. Review Comments to the Author

Reviewer #1: I appreciate the author’s hard work on this study and believe it’s important for this information to be available for the global community. However, the paper still requires significant copy-editing. I have provided some initial suggested edits below, but the full paper would greatly benefit from a full copy edit by someone with experience publishing in an English language journal.

Response: We thank you the reviewer’s constructive comments and suggestions. We copy edited the whole manuscript for typo and grammar errors.

Since you abbreviate trachomatous trichiasis as TT early in the paper, continue using TT throughout rather than spelling out the full name.

Same comment on using POTT throughout.

Response: Thank you for your suggestions. We corrected throughout the document.

Line 56 – rather than stating “…transmitted through contact with eye discharge” I recommend a broader statement, such as “…transmitted through contact with an infected person”. This is because the bacteria could be transmitted via nasal discharge or other fomites. (Trachoma - StatPearls - NCBI Bookshelf (nih.gov))

Response: We corrected it, (line 56).

Line 58 – change “resulting in excruciating pain….” To say something like “causing pain and overtime can result in scarring of the cornea”. The large spectrum of clinical manifestations of TT means that some people do not develop scarring, but many do. So better for this language to reflect this. You all should cite this statement.

Response: Thank you for your suggestions. We edited from the document (line 58).

Line 59 – same comment at line 58 about the large spectrum of clinical manifestations. I would say “If left unmanaged TT can lead to visual impairment and blindness.” You all should cite this statement.

Response: We acknowledge the reviewer’s effort and we edited (line 59).

Line 60 – Trachoma is the leading cause of preventable blindness. I don’t think this is the correct citation. Please double check.

Response: We acknowledge the reviewers concern. We found that this statement is duplicated in the lines 78-80, hence, we merged it and the correct citation is reference no.7 (line 80).

Line 61 – TT can be managed through surgery which involves….(briefly describe the procedure and cite the WHO manual)

Response: We included the types of TT surgery procedures (lines 63-65).

Line 68 – what is “the plan”? since you give a true value “173,945” remove “about”. I think what you are saying is “In Ethiopia, 173,945 received TT surgery and 39,339,311 people received antibiotics through mass drug administration in 2017.”

Response: Thank you. We revised based on the reviewer’s suggestions (line 68).

Line 71 – all villages received health education….

Response: We edited it (line 76).

Line 75/76 – suggest rewording “TT creates economic loss due to reduced productivity resulting form blindness and visual impairment.”

Response: We edited the manuscript based on the reviewer’s suggestions (lines 80-82)

Line 77 – POTT negatively impacts perception and uptake of surgery in communities. Individuals are less likely to seek surgery when they witness POTT in their community. Additionally, re-operation has…..”

Response: We thank you the reviewer’s effort. We edited the statements (lines 83-88).

Line 85 – POTT

Response: Corrected.

Line 88/89 – confusing wording

Response: We reworded the statements (lines95-99).

Line 219 – first time to mention PLTR – I suggest mentioning this approach in the background and you also need to spell out the acronym the first time you use it.

Response: The PLTR has been introduced in the background section of the manuscript (line 65).

Aside from copy-edit. Could the authors please explain how they accounted for quality of original surgery? Were all the surgeons in the study of a certain caliber?

Response: The surgeons were Integrated Eye Care Workers who have had a special training on TT surgery. Except the potential intrinsic individual quality differences they took similar training.

---

## [Decision Letter · Decision Letter 2]

8 May 2024

PONE-D-23-08111R2Postoperative Trachomatous Trichiasis  and Associated Factors Among Adults who underwent Trachomatous Trichiasis Surgery in Ambassel District, North-East Ethiopia.PLOS ONE

Dear Dr. Wassie,

Thank you for submitting your manuscript to PLOS ONE. After careful consideration, we feel that it has merit but does not fully meet PLOS ONE’s publication criteria as it currently stands. Therefore, we invite you to submit a revised version of the manuscript that addresses the points raised during the review process.

In the abstract and other sections, put the AOR values within the brackets.Improve the writing style throughout the document. For instance, the statement “A multi-stage sampling technique was used to employ a total of 506 individuals” could be rephrased.Carefully check typo and punctuation errors throughout. e.g., infections off….. infections of; In the Amhara region 91,977 persons…..In the Amhara region, 91,977 personsThe 4^th^ paragraph of the introduction needs to be moved up (either integrated into the first paragraph or so). You cannot talk about Ethiopia and Amhara region before global reports.  I generally recommend revising the flow of information in the introduction section.Justification needs to be supported with references. E.g., “Previous studies done in Ethiopia were few and limited only in urban areas” include these studies as references.The authors applied correction formula, which was not actually required.Sampling following house number does not seem acceptable as there are scattered households without numbering.In the tables, add the SI units for some variables.I also recommend operational definitions for some variables. e.g., knowledge, type of latrine, toilet utilization.

We look forward to receiving your revised manuscript.

Kind regards,

Mesfin Gebrehiwot Damtew (PhD)

Academic Editor

PLOS ONE

Journal Requirements:

Additional Editor Comments:

Dear authors,

Both reviewers suggest the acceptance of the manuscript. Below find some more comments to improving the document.

• In the abstract and other sections, put the AOR values within the brackets.

• Improve the writing style throughout the document. For instance, the statement “A multi-stage sampling technique was used to employ a total of 506 individuals” could be rephrased.

• Carefully check typo and punctuation errors throughout. e.g., infections off….. infections of; In the Amhara region 91,977 persons…..In the Amhara region, 91,977 persons

• The 4th paragraph of the introduction needs to be moved up (either integrated into the first paragraph or so). You cannot talk about Ethiopia and Amhara region before global reports. I generally recommend revising the flow of information in the introduction section.

• Justification needs to be supported with references. E.g., “Previous studies done in Ethiopia were few and limited only in urban areas” include these studies as references.

• The authors applied correction formula, which was not actually required.

• Sampling following house number does not seem acceptable as there are scattered households without numbering.

• In the tables, add the SI units for some variables.

• I also recommend operational definitions for some variables. e.g., knowledge, type of latrine, toilet utilization.

Reviewers' comments:

Reviewer's Responses to Questions

**Comments to the Author**

1. If the authors have adequately addressed your comments raised in a previous round of review and you feel that this manuscript is now acceptable for publication, you may indicate that here to bypass the “Comments to the Author” section, enter your conflict of interest statement in the “Confidential to Editor” section, and submit your "Accept" recommendation.

Reviewer #2: All comments have been addressed

Reviewer #3: All comments have been addressed

2. Is the manuscript technically sound, and do the data support the conclusions?

Reviewer #2: Yes

Reviewer #3: Yes

3. Has the statistical analysis been performed appropriately and rigorously? 

Reviewer #2: N/A

Reviewer #3: Yes

4. Have the authors made all data underlying the findings in their manuscript fully available?

Reviewer #2: (No Response)

Reviewer #3: Yes

5. Is the manuscript presented in an intelligible fashion and written in standard English?

Reviewer #2: Yes

Reviewer #3: Yes

6. Review Comments to the Author

Reviewer #2: Postoperative Trachomatous Trichiasis and Associated Factors Among Adults who underwent Trachomatous Trichiasis Surgery in Ambassel District, North-East Ethiopia.

Authors have incorporated all the correction suggested by reviewer. Now it may be accepted

Reviewer #3: The paper entitled ‘Postoperative Trachomatous Trichiasis and Associated Factors Among Adults who underwent Trachomatous Trichiasis Surgery in Ambassel District, North-East Ethiopia’ The title is very important and it is tries to address the current hot topics and the study makes significant contributions to the study area and beyond.

Line 75: ‘POTT negatively impacts perception and uptake of surgery in communities.’ A sentence cannot start by an abbreviation/acronym, so please write the long form of ‘POTT’.

Lines 145 &146: Include the Institutional Review Board (IRB) Committee decision reference number

Line 172: Editorial issue; ‘The Prevalence of Postoperative Trachomatous Trichiasis’. Here ‘Trachomatous’ should be written in bold.

Line 175: ‘Factors associated with postoperative Trachomatous Trichiasis’. Here also ‘Trachomatous’ should be written in bold.

7. PLOS authors have the option to publish the peer review history of their article (what does this mean?). If published, this will include your full peer review and any attached files.

Reviewer #2: **Yes: **Tabarak Malik

Reviewer #3: No

---

## [Author Response · Author response to Decision Letter 2]

10 May 2024

Response to the Reviewers

Dear editor, thank you for considering our manuscript for publication; we have revised the contents based on the reviewer’s comments as below.

Ref: Submission ID: [PONE-D-23-08111R2] - [EMID:07797ed50f9ecbe2] .

Title: Postoperative Trachomatous Trichiasis and Associated Factors among Adults who underwent Trachomatous Trichiasis Surgery in Ambassel District, North-East Ethiopia.

PLOS ONE

Additional Editor Comments:

Dear authors,

Both reviewers suggest the acceptance of the manuscript. Below find some more comments to improving the document.

1. In the abstract and other sections, put the AOR values within the brackets.

Response: Revised as the editors suggestions (lines 37-43).

2. Improve the writing style throughout the document. For instance, the statement “A multi-stage sampling technique was used to employ a total of 506 individuals” could be rephrased.

Response: We revised the whole document for formatting and typo errors.

3. Carefully check typo and punctuation errors throughout. e.g., infections off….. infections of; In the Amhara region 91,977 persons…..In the Amhara region, 91,977 persons

Response: We revised the whole document for formatting and typo errors.

4. The 4th paragraph of the introduction needs to be moved up (either integrated into the first paragraph or so). You cannot talk about Ethiopia and Amhara region before global reports. I generally recommend revising the flow of information in the introduction section.

Response: Thank you for your comment. We moved up 4th paragraph to the 3rd paragraph place (lines 68-71).

5. Justification needs to be supported with references. E.g., “Previous studies done in Ethiopia were few and limited only in urban areas” include these studies as references.

Response: we included references14 and 15.

6. The authors applied correction formula, which was not actually required.

Response: Correction formula was applied as a rule of thumb since the source population was small, that is 2133.

7. Sampling following house number does not seem acceptable as there are scattered households without numbering.

Response: In this time, there is the so called community health information system (CHIS) or family folder run in the Ethiopian health extension program; in which each household assigned a specific identification number. Hence, study participants were tracked through this system.

8. In the tables, add the SI units for some variables.

Response: we revised as the editors suggestions.

9. I also recommend operational definitions for some variables. e.g., knowledge, type of latrine, toilet utilization.

Response: We have operationalized some variables as the editor recommended (lines 118-122).

Reviewers' comments:

Reviewer #2: Postoperative Trachomatous Trichiasis and Associated Factors Among Adults who underwent Trachomatous Trichiasis Surgery in Ambassel District, North-East Ethiopia.

Authors have incorporated all the correction suggested by reviewer. Now it may be accepted

Response: Thank you for your considerations.

Reviewer #3: The paper entitled ‘Postoperative Trachomatous Trichiasis and Associated Factors Among Adults who underwent Trachomatous Trichiasis Surgery in Ambassel District, North-East Ethiopia’ The title is very important and it is tries to address the current hot topics and the study makes significant contributions to the study area and beyond.

10. Line 75: ‘POTT negatively impacts perception and uptake of surgery in communities.’ A sentence cannot start by an abbreviation/acronym, so please write the long form of ‘POTT’.

Response: Revised as the reviewer’s suggestion (line, 75).

11. Lines 145 &146: Include the Institutional Review Board (IRB) Committee decision reference number

Response: We included the IRB committee protocol number (line 146).

12. Line 172: Editorial issue; ‘The Prevalence of Postoperative Trachomatous Trichiasis’. Here ‘Trachomatous’ should be written in bold.

Response: Thank you for your reminder: we have edited it (line 172).

13. Line 175: ‘Factors associated with postoperative Trachomatous Trichiasis’. Here also ‘Trachomatous’ should be written in bold.

Response: Thank you for your reminder: we have edited it (line 175).

---

## [Editor Report · Decision Letter 3]

13 May 2024

Postoperative Trachomatous Trichiasis  and Associated Factors Among Adults who underwent Trachomatous Trichiasis Surgery in Ambassel District, North-East Ethiopia.

PONE-D-23-08111R3

Dear Dr. Wassie,

We’re pleased to inform you that your manuscript has been judged scientifically suitable for publication and will be formally accepted for publication once it meets all outstanding technical requirements.

Kind regards,

Mesfin Gebrehiwot Damtew (PhD)

Academic Editor

PLOS ONE
---

## [Editor Report · Acceptance letter]

16 May 2024

PONE-D-23-08111R3 

PLOS ONE

Dear Dr. Wassie, 

I'm pleased to inform you that your manuscript has been deemed suitable for publication in PLOS ONE. Congratulations! Your manuscript is now being handed over to our production team.

Kind regards, 

on behalf of

Dr. Mesfin Gebrehiwot Damtew 

Academic Editor

PLOS ONE